# Association of Maternal Smoking during Pregnancy with Neurophysiological and ADHD-Related Outcomes in School-Aged Children

**DOI:** 10.3390/ijerph20064716

**Published:** 2023-03-07

**Authors:** Karina Jansone, Anna Eichler, Peter A. Fasching, Johannes Kornhuber, Anna Kaiser, Sabina Millenet, Tobias Banaschewski, Frauke Nees

**Affiliations:** 1Department of Child and Adolescent Psychiatry and Psychotherapy, Medical Faculty Mannheim, Central Institute of Mental Health, University of Heidelberg, 68159 Mannheim, Germany; 2Department of Child and Adolescent Mental Health, University Hospital Erlangen, Friedrich-Alexander University Erlangen-Nürnberg, 91054 Erlangen, Germany; 3Department of Obstetrics and Gynecology, University Hospital Erlangen, Friedrich-Alexander University Erlangen-Nürnberg, 91054 Erlangen, Germany; 4Department of Psychiatry and Psychotherapy, University Hospital Erlangen, Friedrich-Alexander University Erlangen-Nürnberg, 91054 Erlangen, Germany; 5Institute of Medical Psychology and Medical Sociology, University Medical Center Schleswig-Holstein, Kiel University, 24105 Kiel, Germany

**Keywords:** EEG, maternal smoking, pregnancy, ADHD, hyperactivity, resting-state brain activity

## Abstract

Data of a longitudinal cohort study were analyzed to investigate the association between prenatal tobacco exposure and electroencephalographical (EEG) power spectrum in healthy, school-aged children as well as its relationship with attention deficit hyperactivity disorder (ADHD)-related symptoms. Group comparisons (exposed, non-exposed) were performed to test whether prenatal tobacco exposure was associated with brain activity and ADHD symptoms, with adjustments made for covariates including child’s sex, child’s age, maternal age, maternal smoking habit before pregnancy, alcohol consumption during pregnancy, gestation age, and maternal psychopathology. Tobacco-exposed children showed higher brain activity in the delta and theta frequency bands. This effect was independent of the considered covariates. However, the effects on hyperactivity were found to significantly depend on maternal age and alcohol consumption during pregnancy, but not on the amount of exposure. In summary, smoking during pregnancy significantly affected the resting-state brain activity in children, independent of socio-demographic factors, indicating potential long-lasting effects on brain development. Its impact on ADHD-related behavior was shown to be influenced by socio-demographic confounding factors, such as maternal alcohol consumption and the age of the mother.

## 1. Introduction

Pregnancy is a period that is vital for the physiological and psychological development of a child. Health-related risk behaviors of the mother during this period can therefore have highly relevant consequences for the child [1,2]. A growing body of literature over the past decades has provided evidence that specifically, maternal smoking during pregnancy results in an increased risk of adverse neurodevelopmental and health outcomes during childhood and adolescence (e.g., [3,4]). 

Tobacco smoke contains thousands of known toxic components including nicotine and carbon monoxide. These can cause early life changes at the level of enzymes and hormones as well as on the expression of genes, micro RNAs, and proteins in the child [5], and are potential mechanisms underlying an increased risk of adverse outcomes observed in children exposed to maternal smoking during pregnancy. Prenatal smoking was shown to lead to deleterious effects on cognitive [6] and neurobehavioral developmental processes [7] during childhood including externalizing behaviors of oppositional defiant problems, hyperactivity, inattention [8], and impairments in intellectual functioning associated with auditory processing, reading, and language development [9]. In this respect, several review articles [10,11] have also described an association between maternal smoking and diagnoses of mental disorders, pointing to higher rates of conduct disorder and ADHD in prenatally tobacco-exposed compared to non-exposed children (e.g., [12,13,14]). Nevertheless, some studies suggest that exposure to maternal cigarette smoking in pregnancy might only be indirectly linked to ADHD symptoms, and rather reflect an interaction with factors related to socio-environmental load such as family-related ones of parental psychopathology including an additional involvement of genetic factors in the case of parental ADHD [15]. Another study by Langley et al. (2011) compared the risks of maternal and paternal smoking during pregnancy on offspring ADHD. The study demonstrated that ADHD symptoms were significantly associated with exposure to both maternal and paternal smoking during pregnancy. Even when paternal smoking was examined in the absence of maternal smoking, the associations remained significant [16]. Moreover, paternal smoking was still significant after controlling for genetic and household-level factors.

It should be noted that previous research has demonstrated the impact of prenatal smoking exposure on brain development. For instance, an example of such effects can be seen in studies that have found that exposure to prenatal tobacco can lead to a reduction in brain volume including regional thinning in areas such as the superior frontal, superior parietal, lateral occipital, and precentral cortices [17,18], white matter [19], and subcortical regions such as the amygdala, cerebellum, and the corpus callosum in the brains of newborns [20]. These structural changes were also observed to be accompanied by functional and behavioral alterations, for instance, changes in the cerebellum were found to relate to emotional, impulse control, and attentional processes [21]. 

Importantly, these early changes in the brain and alterations in behavior and psychopathology might persist into adolescence and adulthood. For example, Holz et al. [22] showed that maternal smoking is associated with changes in brain regions related to behavioral response inhibition, together with ADHD symptomatology. The results of the study indicated an inverse relationship between inferior frontal gyrus activity and ADHD symptoms in prenatally tobacco-exposed children. 

Aside from fMRI, some studies on maternal smoking during pregnancy also used electroencephalography (EEG) as a measure of brain functions [23]. Due to its non-invasive nature and high millisecond temporal resolution, EEG represents a specifically valid measure not only with respect to brain development [24], but also as an indicator of behavioral control [25]. The few available data on EEG and maternal smoking during pregnancy stem from investigations during childhood. In these studies, a consistent and synchronized pattern of activity across different brain regions involved in efficient communication and information processing has been found. In studies that investigated individuals with ADHD, this pattern of brain activity was shown to be disrupted and less synchronization between regions was found. In a prospective cohort study, Shuffrey et al. (2020) [26] found significantly increased right-central beta (19–24 Hz), low gamma (28–36 Hz), and increased right-parietal low gamma (28–36 Hz) and high gamma (37–45 Hz) EEG power in prenatally tobacco-exposed neonates compared to non-exposed infants [26]. Such negative effects of fetal exposure to parental tobacco smoking might further cumulate negatively with smoking during lactation and with second-hand smoking exposure [2]. Investigations of the influence of prenatal smoking on electrical brain activity, assessed with EEG, in the offspring are therefore limited so far, and did not investigate the effects into childhood. 

Nevertheless, previous studies have indicated associations between behavioral symptoms of ADHD with EEG power spectra [27,28,29]. School-aged children diagnosed with ADHD showed increases in the theta activity and the theta/beta ratio, increased frontal delta, reduced global alpha and frontal beta activity during an eyes-closed resting condition [30] compared to healthy controls. Frontal theta activity correlated with inattention, while the theta/beta ratio correlated with hyperactivity-impulsivity. Similarly, the results of the study of Rodríguez-Martínez et al. observed an increase in the delta power and decreased beta power in children and adolescents with ADHD compared to control subjects [31]. 

In the present study, we aimed to (a) examine the influence of prenatal smoking on EEG brain activity in school-aged children, taking into consideration the amount of smoked cigarettes during pregnancy and (b) analyze its relations to ADHD-associated behaviors. For ADHD, particularly higher rates in hyperactive and impulsive behavior might be related to prenatal smoking as indicated by earlier work [14]. Moreover, ADHD is also characterized by functional changes in brain regions that were found to be affected by prenatal tobacco exposure in the newborn brain, including reduced delta power in the fronto-central and parietal regions [26]. Therefore, we also hypothesize the main changes in these brain regions in our adolescent sample. Because of previous findings [15,16,32], maternal psychopathology and alcohol consumption during pregnancy as well as the child’s sex, child’s age, maternal age, and maternal smoking habit before pregnancy and gestation age were included in the analysis as additional cofounders in order to investigate the true risk effect of prenatal tobacco exposure. 

## 2. Materials and Methods

We used data from the longitudinal Franconian Maternal Health Evaluation Studies (FRAMES, Erlangen, Bavaria, Germany; see [33]) and the follow-up Franconian Cognition and Emotion Studies (FRANCES, Erlangen, Bavaria, Germany [34]). The FRAMES data were obtained between the years 2005 and 2007 and the total sample on the baseline assessments consisted of 1100 pregnant women older than 18 years. These women had all reached a gestational age of at least 30 full weeks. Perinatal maternal health data were collected at the Department of Obstetrics and Gynecology [33,35]. 

Between the years 2012 and 2015, when the children attended primary school, 618 women were contacted again via telephone for participation in the follow up FRANCES assessments. Finally, n = 245 FRAMES mother–child dyads (39.6%; child age: M = 7.74, SD = 0.74, range 6.00–9.90) agreed to take part in the FRANCES I follow-up wave. These assessments took place at the Department of Child and Adolescent Mental Health in Erlangen, Germany, where the mothers filled out the questionnaires and children were tested for cognitive abilities and the EEG measurements were performed [36]. The studies are consistent with the Declaration of Helsinki and were approved by the Local Ethics Committee of the University Hospital Erlangen (no. 3374 (FRAMES) and 4596 (FRANCES)). Before participation, all subjects received detailed information about the study and gave their written informed consent.

### 2.1. Assessment of Maternal Prenatal Smoking 

Maternal smoking behavior was evaluated at two time points, first in the FRAMES study and then in the FRANCES study. In the FRAMES study, the maternal smoking behavior was assessed through a gestational questionnaire (face-to-face interview), where women were asked to self-report their smoking behavior during pregnancy. The interview consisted of a series of questions related to the frequency and amount of cigarettes smoked by the women during pregnancy. In the FRANCES study, maternal prenatal smoking was assessed through a retrospective self-report screening questionnaire. This questionnaire included questions about the number of cigarettes smoked per week during each of the three trimesters of pregnancy, as well as before pregnancy and during lactation. The study also gathered information on the father’s smoking behavior and its potential impact on the mother’s exposure to secondhand smoke during pregnancy. Overall, the smoking assessments in both studies provide a comprehensive picture of the maternal smoking behavior over time, and the responses regarding smoking during pregnancy were consistent across the overlapping items from the face-to-face-interview and the retrospective questionnaire.

### 2.2. Assessment and Pre-Processing of Resting-State EEG Activity

Children completed 2.5 min of eyes-open and 2.5 min of eyes-closed resting-state EEG. During the eyes-open resting state, participants fixated on a point in front of them and were encouraged to minimize both ocular and other bodily movements. EEG activity was recorded from 25 sites (10–20 system plus additional midline electrodes and mastoid electrodes; recording reference: Fcz, ground electrode: CP2), with standard electrode caps with sintered Ag/AgCl electrodes (Easycap, Herrsching, Germany). The raw EEG data were inspected, pre-processed, and analyzed offline using a BrainVision Analyzer (Version 2.2.0). Filter bandwidth was set to 0.016–120 Hz; the sampling frequency was 500 Hz. The resistance of the electrodes was kept below 20 kΩ. The EEG data were pre-processed using several steps including downsampling to 250 Hz, filtering, and the removal of artifacts caused by eye blinking, heart beating, muscle movements, or loose/broken electrodes. To ensure more sensitive and reliable analyses, rejection techniques such as filtering were applied. The low pass filter was set to 0.1 Hz to capture the basic cortical rhythms underlying higher brain functions, while the high pass filter was set to 70 Hz. In order to attenuate artifacts caused by external devices such as the electrical power supply of devices in the recording room, we used a notch filter of around 50 Hz. After artifact rejection, frequency analysis was performed in order to explore the EEG data gathered during the two resting state conditions of “eyes open” and “eyes closed”. Ocular (blinks and saccades) and any other remaining artifacts such as muscular or cardiac effects were isolated and rejected via independent component analysis (ICA) on the continuous data. Components for rejection were selected manually. Using a Hanning window with a 10% taper length, fast Fourier transformations were conducted with non-overlapping 2.048 s epochs of corrected data. By using a built-in algorithm in Brain Vision Analyzer, fast Fourier transform (FFT) was applied to transform the time–domain EEG epochs into equivalent frequency–domain epochs. Afterward, the obtained FFT values of delta, theta, alpha, and beta were extracted using the FFT band export option. After artifact detection and rejection, the averaged data consisted of 74% of good segments from the eyes-open resting-state condition and 68% of the good segments from the eyes-closed condition.

Finally, we averaged the participants’ data individually across the epochs for each electrode site and for the frequency bands (delta, theta, alpha, beta) and the mean absolute power was computed and exported in the form of text files. Participants were only included if their EEG data contained at least 50% of artifact-free segments. The recorded activity was divided into three regions using an average value for each region. The posterior region represented the averaged activity in electrodes T5, P3, O1, P2, T6, P4, O2; the frontal region averaged activity from Fp1, Fp2, F3, F7, FCz, Fz, F4, F8, and the average of the T3, C3, Cz, T4, and C4 electrodes was used as an indicator of the central area. For both resting-state conditions, “eyes open” and “eyes closed”, the absolute power was obtained for four frequency bands: theta (0.5–3.5 Hz), delta (3.5–7.5 Hz), alpha (7.5–12.5 Hz), and beta (12.5–30 Hz). 

### 2.3. Assessment of ADHD-Related Behavioral Difficulties 

For behavioral difficulties, we referred to the German screening instrument Parent-assessment ADHD (DISPYPS-II, Fremdbeurteilungsbogen ADHS, FBB-ADHD [37]), a 20-item questionnaire (here: mother-report) that captures the following three dimensions: inattention, impulsivity, and hyperactivity. It includes an assessment of the severity and perceived burden of each dimension and the sum of all dimensions represents the total score of ADHD. 

### 2.4. Statistical Analysis 

The statistical analyses were performed using two types of software: SPSS 27.0 (IBM SPSS Statistics, IBM Corporation, Armonk, NY, USA) and R. In R, the “mgcv” package (Version 1.8–28) was utilized, which is specifically designed for generalized additive modeling (GAM) [38,39]. The “mgcv” package includes GAM using a variety of smoothing functions such as cubic splines or penalized regression splines, and allowed for a more flexible and nuanced analysis of the present data, taking into account the potential nonlinear effects of the predictor variables on the EEG measures. In the first step, we performed analyses for the sample characteristics with respect to smoking-exposed vs. non-exposed children including correlations between our outcome variables (EEG power spectrum and ADHD-related symptoms). For the main hypotheses, we then examined (a) the effects of prenatal smoking as a dichotomous variable (prenatally tobacco exposed vs. unexposed) on the EEG power spectrum and FBB-ADHD scales of impulsivity, attention deficit, and hyperactivity, and (b) in the children who experienced prenatal tobacco exposure, we explored the linear and nonlinear associations between the estimated averaged amount of weekly smoked cigarettes of the mother during pregnancy and brain frequencies as well as the ADHD-related outcomes. We used a series of generalized additive mixed models (GAMMs) [40,41], adjusting for fixed and random effects (spline models) [42]. Here, 1.5% winsorization was applied to convert extreme outliers. This approach follows prior studies on the effects of prenatal alcohol exposure on the psychological, behavioral, and neurodevelopmental outcomes in children (e.g., [43]). For all analyses, we ran covariate-unadjusted and -adjusted models, using the following covariates for the adjusted models: child’s sex, child’s age, maternal age at child birth or FRANCES assessment, maternal psychopathology (interview question with dichotomous self-rating, FRANCES assessment), maternal smoking before pregnancy, maternal alcohol drinking during pregnancy (assessed retrospective via interview with dichotomous self-rating in the third trimester), and week of pregnancy at birth.

## 3. Results

The FRANCES cohort included 248 parents and children each. For our data analyses, we only used full datasets, which resulted in a final sample of N = 142 parents and children each. The mothers aged 19 to 41 (*M* = 32.68, *SD* = 4.32) at the child’s birth, fathers aged 23 to 52 (*M* = 35.25, *SD* = 5.50) at the child’s birth, and children aged 6 to 9 years (*M* = 7.73, *SD* = 0.67) at FRANCES assessment. Subject characteristics were described as frequency, mean, and standard deviation (see Table 1). None of the children had a history of clinically significant developmental or intellectual disorders or clinically significant somatic abnormalities. The information on parental psychopathology was obtained with trained psychologists. According to the maternal responses, 34 mothers reported the presence of at least one psychiatric diagnosis. 

For the mean values of the maternal smoking behavior and the number of weekly smoked cigarettes for active and passive smoking (for illustration see Table 2), we found that, in total, 26 women smoked during pregnancy and 116 women were in the non-smoker group. Thereby, 35 women from the non-smoker group reported passive smoking during pregnancy. For the main data analyses, we used the mean value of three obtained values from each trimester with respect to the averaged quantity of cigarettes smoked per week during pregnancy.

### 3.1. Effects of Prenatal Tobacco Exposure on Brain Activity 

**Exposure vs. non-exposure.** For the covariate-unadjusted models, we found significant effects of maternal smoking during pregnancy on the delta central, posterior and overall in the frontal, posterior, and overall theta as well as on the frontal, central, and overall alpha brain activity in the eyes-closed resting-state condition (see Appendix A in the online supplement). Here, the prenatally exposed group obtained significantly higher brain activity. In the eyes-open resting-state condition, significant effects were shown in the frontal, central, posterior, and overall theta as well as in the central alpha (see Appendix A in the online Appendix A) brain activity, with prenatally tobacco-exposed children compared to non-exposed children also showing significantly higher brain activity in these areas. When adjusting for the covariates (child’s sex, child’s age, maternal age; maternal psychopathology; maternal smoking before pregnancy; maternal alcohol drinking, week of pregnancy at birth), the effects for the delta and theta frequency bands in both the eyes-open and eyes-closed conditions remained significant (see forest plot in Figure 1; for the effects of the included covariates, see Appendix A in the online Appendix A). 

**Linear and nonlinear associations.** For the covariate-unadjusted models, we found both linear and nonlinear associations between the estimated averaged number of weekly smoked cigarettes during pregnancy and brain activity, with linear associations for the delta central, and nonlinear associations for the theta posterior, theta central, theta overall, and delta posterior. These associations remained significant when adjusting for the covariates (see spline models in Figure 2).

### 3.2. Effects of Prenatal Tobacco Exposure on ADHD

Exposure vs. non-exposure. For the unadjusted models, we observed a significant effect of maternal smoking during pregnancy on the hyperactivity scores of the children where prenatally exposed children (*M =* 5.33, *SD =* 1.45) showed higher rates of hyperactivity compared to prenatally unexposed (*M =* 4.75, *SD* = 2.28) children (see Appendix A in the online Appendix A). No significant effects were observed for impulsivity (exposed: *M* = 5.15, *SD* = 2.49; unexposed: *M* = 5.14, *SD* = 2.05), attention deficits (exposed: *M* = 5.85, *SD* = 1.41; unexposed: *M* = 5.72; *SD* = 1.59), and the total FBB-ADHD score (exposed: *M* = 5.58, *SD* = 1.38; unexposed: *M* = 5.19, *SD* = 1.65). When adjusting for the covariates, the effect on hyperactivity did not survive (see Appendix A in the online Appendix A).

Linear and nonlinear associations. For both the covariate-adjusted and -unadjusted models, we did not observe any significant associations between the estimated number of smoked cigarettes of the mother and ADHD-related symptoms.

### 3.3. Interaction of Prenatal Tobacco Exposure, Brain Activity, and ADHD Symptoms

With respect to the associations between the outcome variables, brain activity, and ADHD-related symptoms, in the eyes-open condition, we found significant partial correlations between impulsivity and frontal (r = 0.21, *p* < 0.05), central (r = 0.22, *p* < 0.05) alpha brain activity, central (r = 0.31, *p* < 0.05), frontal (r = 0.20, *p* < 0.05), posterior (r = 0.20, *p* < 0.05), overall delta (r = 0.32, *p* < 0.05) frequency power, and frontal (r = 0.21, *p* < 0.05) theta brain activity. Hyperactivity and the total ADHD score were significantly associated with frontal (r = 0.25, *p* < 0.01; r = 0.22, *p* < 0.01), central (r = 0.33, *p* < 0.01; r = 0.31, *p* < 0.01), posterior (r = 0.23, *p* < 0.01; r = 0.21, *p* < 0.01), and overall (r = 0.34, *p* < 0.01; r = 0.30, *p* < 0.01) delta brain activity. No significant correlation was observed between attention deficit and any of the frequencies in the eyes-open condition. In the eyes-closed condition, significant correlations were observed between the central delta frequency power and impulsivity (r = 0.22, *p* < 0.05), hyperactivity (r = 0.22, *p* < 0.05), and the total ADHD score (r = 0.20, *p* < 0.05). Posterior delta frequency power was also significantly associated with the total ADHD score (r = 0.21, *p* < 0.05). As for the eyes open condition, no significant correlation was observed between attention deficit and any of the frequencies in the eyes-closed condition (see Appendix A in the online Appendix A).

Since brain activity in the alpha, delta, and theta frequencies was positively associated with ADHD-related symptoms (hyperactivity, impulsivity, total ADHD score), mediation analysis was performed in order to test whether there were brain-activity-mediated associations between prenatal maternal smoking and behavioral outcomes (FBB-ADHD scales) in the covariate-adjusted models. Here, differential mediation was observed, where prenatal smoking was significantly associated with brain activity (alpha, delta, and theta frequencies), and brain activity (alpha and delta frequencies) was significantly associated with impulsivity, whereas brain activity did not mediate the relationship between prenatal smoking and ADHD-related behavior (see Appendix A in the online Appendix A).

## 4. Discussion

In the present study, we aimed to investigate the influence of prenatal smoking on (a) EEG brain activity, (b) ADHD-related symptoms in school-aged children, and (c) their interaction, taking into account the number of smoked cigarettes during pregnancy as well as considering potential confounding factors including the child’s sex, child’s age, maternal age at the child’s birth, maternal psychopathology, maternal smoking before pregnancy, maternal alcohol drinking during pregnancy, and week of pregnancy at birth.

Previous studies have shown that maternal smoking during pregnancy is a major prenatal risk factor for child development: excessive prenatal tobacco exposure has been related to adverse behavioral outcomes including hyperactivity and impulsivity to deficits in cognitive abilities such as auditory and visual attention performance accuracy, and to changes in the brain’s structure such as a significant reduction in cortical gray matter in young children [11,44,45]. With respect to changes in the brain, the majority of these studies has so far focused either on fetal structural and functional brain development or on brain correlates in newborns [26,45,46,47], mainly using MRI [22,44,48]. Moreover, studies have so far been partly restricted by relatively small sample sizes [26,46,47], and proximal risk factors such as socio-demographical characteristics may have co-determined or masked effects (e.g., [49]).

Capitalizing on the EEG, behavioral, and clinical data with socio-demographical relations such as maternal psychopathology from a larger sample of mothers and their children, we found significant changes in the resting-state EEG as a response to prenatal tobacco, with increased delta and theta brain activity in the resting-state EEG in school-aged exposed compared to non-exposed children. Brain activity in these frequency bands was also significantly related to the number of cigarettes smoked. This is in line with the previously above-mentioned studies that reported an association between prenatal tobacco exposure and changes in the brain of newborns and young children [11,44,45]. In our study, we also found that these effects were independent of potential confounding covariates—both the covariate-unadjusted and -adjusted models were significant. This adds information to the previous literature highlighting that there might be some single specific effects that are rather independent of proximal risk factors. Furthermore, we explored the effects of prenatal maternal smoking on the behavior and clinical symptoms. For the ADHD-related symptoms, the effects of prenatal maternal smoking were only significant in the unadjusted models including potential confounding factors; when controlling for covariates, the effects did not remain significant. This indicates the need to take into account the proximal risk factors, particularly when it comes to the effects of prenatal tobacco exposure on the behavior and clinical symptoms. Some previous studies have also made this point, highlighting that the association of ADHD with prenatal tobacco exposure symptoms might not be directly affected, but might rather covary with intra-household-related factors [15,16]. Rice et al. (2018) stated that the negative causal link between maternal smoking and physiological characteristics such as child birth weight seemed to be a rather direct effect [50], consistent across various study designs [16,51]; however, when it comes to children’s psychopathology, a broader set of risk factors including socioeconomic status (e.g., income conditions), parental psychopathology, maternal stress [50], and maternal age [52] come into play. This was also indicated by a large prospective birth cohort study, where the association between prenatal tobacco exposure and offspring ADHD was found to be significantly related to the confounding factors. The association may therefore not necessarily be indicative of causal intrauterine effects [53].

These and our findings raise several questions for future studies, for example, whether or to which degree these are divergent pathways, and at which point during development factors might strongly interact. Associations between prenatal smoking exposure during pregnancy and behavior problems might not be simply direct or causal (see, e.g., [15]). 

In the present study, we also observed significant partial correlations between impulsivity and hyperactivity and frontal alpha (for impulsivity only) as well as frontal, central, posterior, and overall delta frequencies (also for the total ADHD score). Such associations have also been previously reported, for example, increased alpha and theta activity [27,54,55] as well as increased delta activity [54] has been found in children with an ADHD diagnosis compared to typically developing children. This emphasizes that the developmental pathway of prenatal tobacco exposure is complex and involves not only brain–behavior interactions, but also socio-demographic and psychosocial factors. Since the present study involved typically developing children without an ADHD diagnosis, this might have an effect, even at lower rates of ADHD symptoms. 

The present study needs to be seen in the light of some limitations. First, we assessed smoking behavior during pregnancy retrospectively with a self-report questionnaire. This could have resulted in a bias, for example, in that parents might underreport smoking during pregnancy due to social desirability. However, double checking the women’s answers regarding their smoking behavior in both the FRAMES and FRANCES assessments provides a reliable measure of their smoking behavior over time. In particular, the fact that the answers were consistent across the two assessments increases the confidence in the validity of the self-reported data. This helps to ensure that the data accurately reflect the smoking behavior of the women during pregnancy and after childbirth. Second, the ADHD scores in our sample were rather low. While on one hand one could argue that this may dampen the generalizability of the findings, on the other hand, we could provide added insights into a low-risk sample, which are also interesting with respect to brain-related associations. Additionally, with only 26 children exposed to prenatal tobacco, the study may be underpowered to detect significant effects on ADHD-related symptoms. While our findings suggest that prenatal tobacco exposure was not significantly associated with hyperactivity in this sample, larger studies may be needed to confirm this result and explore potential interactions with other covariates.

## 5. Conclusions

In summary, to the best of our knowledge, this is one of the first studies that examined the effects of maternal smoking during pregnancy on brain activity assessed with EEG and behavioral data. While smoking during pregnancy had an impact on the resting-state brain activity in children, which persisted irrespective of socio-demographic factors and hence may indicate long-lasting effects on brain development, the effects on ADHD-related behavior appeared to significantly depend on socio-demographic confounding factors such as maternal alcohol consumption and age of the mother. Future research should aim to investigate the effects of maternal smoking on the oscillatory power in offspring over different time points to better understand the temporal dynamics of the effects and thus enable the characterization of developmental changes in disorder specificity in EEG profiles. Additionally, future studies should consider different ADHD symptom scores and ADHD diagnosis longitudinally. Furthermore, it is crucial to disentangle and integrate socio-demographic and cultural circumstances in a more comprehensive manner.

## Figures and Tables

**Figure 1 ijerph-20-04716-f001:**
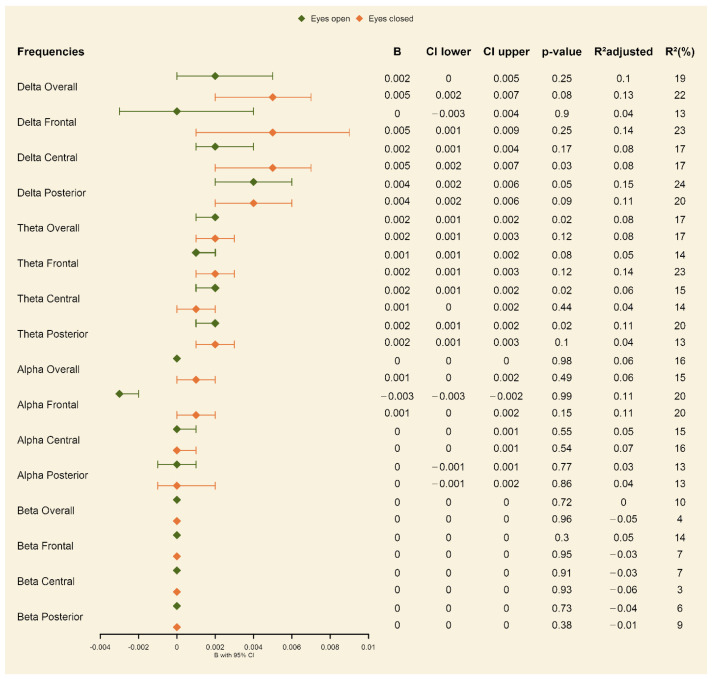
Forest plot showing the association between smoking during pregnancy and offspring EEG activity when comparing the exposed and non-exposed groups, adjusted for covariates (child’s sex, child’s age, maternal age, maternal psychopathology, maternal smoking before pregnancy, maternal alcohol drinking, week of pregnancy at birth).

**Figure 2 ijerph-20-04716-f002:**
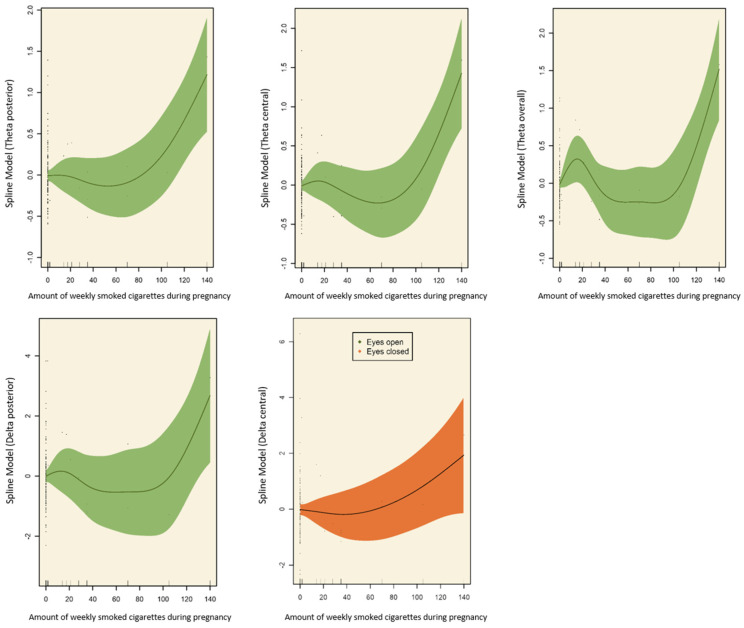
Spline models demonstrating a significant relationship between the estimated number of smoked cigarettes of the mother during pregnancy and offspring brain activity (delta and theta frequency bands), adjusted for covariates (child’s sex, child’s age, maternal age, maternal psychopathology, maternal smoking before pregnancy, maternal alcohol drinking, week of pregnancy at birth).

**Table 1 ijerph-20-04716-t001:** Sample characteristics with children divided into two groups of the tobacco non-exposed and prenatal tobacco-exposed children.

*Characteristic*	Tobacco Unexposed Children (n = 116)		Children With Prenatal Tobacco Exposure (n = 26)		
**Youth Variables**					
	*n*	%	*n*	%	*p*
Sex					0.434
Male Female	59 57	50.9 49.1	11 15	42.3 57.7	
Prenatal alcohol exposure					0.562
No Yes	95 21	81.9 18.1	20 6	76.9 23.1	
	Mean	SD	Mean	SD	*p*
Age (years)	7.72	0.65	7.76	0.74	0.781
Gestational age	39.41	1.33	39.23	1.45	0.533
Birth weight	3453.91	494.28	3370.77	520.34	0.446
Intelligence quotient *	104.23	10.25	104.31	7.18	0.972
**Parent Variables**					
Age at giving birth	32.86	4.12	31.88	5.15	0.299
	*n*	%	*n*	%	*p*
Prenatal marital status					0.068
Married Single parent	113 3	97.4 2.6	23 3	88.5 11.5	
Maternal Psychopathology					0.536
No Yes Unknown	68 29 19	58.6 25.0 16.4	17 5 4	65.4 19.2 15.3	
Paternal Psychopathology					0.461
No Yes Unknown	92 12 12	79.3 10.3 8.3	20 2 4	76.9 7.7 15.3	

* Intelligence quotient was assessed with the Intelligence and Development Scales (IDS) test battery (Grob, Meyer, 2013)**.**

**Table 2 ijerph-20-04716-t002:** Detailed information on maternal smoking behavior.

	Smoked	Did Not Smoke	Passive Smoking
	*n* (%) *[Min, Max]* *M (SD)*	*n* (%)	*n* (%) *[Min, Max]* *M (SD)*
Before pregnancy	40 (28.2) [2, 300] 25.21 (50.72)	102 (71.8)	33 (32.4) [3, 210] 102.39 (57.44)
1. Trimester	26 (18.3) [2, 140] 5.25 (19.28)	116 (81.7)	40 (34.5) [4, 210] 98.16 (57.09)
2. Trimester	24 (16.9) [2, 140] 5.07 (19.25)	118 (83.1)	37 (31.4) [4, 210] 99.05 (56.39)
3. Trimester	24 (16.9) [2, 140] 4.83 (19.01)	118 (83.1)	35 (29.7) [4, 210] 99.00 (56.48)
During pregnancy	26 (18.3) [2, 140] 5.09 (19.23)	116 (81.7)	35 (30.2) [4, 210] 100.51 (55.27)

## Data Availability

Due to ethical, legal, or privacy issues are present, the data should not be shared.

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
