# Peer review of "Association of Maternal Smoking during Pregnancy with Neurophysiological and ADHD-Related Outcomes in School-Aged Children"

_ijerph, 2023, doi:10.3390/ijerph20064716_

Round 1

Reviewer 1 Report

The present manuscript well documents the covariates-independent association of maternal smoking during pregnancy with neurophysiological outcomes.  but did not found a covariates-independent association of maternal smoking during pregnancy with ADHD-related outcomes in school aged children. With respect to the description of results to ADHA-related outcomes there should be made, however, some adjustments throughout the manuscript to more specifically describe the main findings.

Minor comments

To abstract/results

In 3.2 you show that with adjusted data prenatal tobacco-exposure has no effect on hyperactivity. However, in the abstract this main result has not been highlighted as such. Please describe this effect properly.  

Regarding the significant role of the two covariates maternal age and alcohol consumption for hyperactivity: Where you present data for that? Could you additionally specify the direction of these effects?

To Tables

It seems that Figure 1 presents main data repeatedly given in S1! Please avoid such redundancies.

Missing are data for ADHD-related symptoms.

To 2.4 Statistical analysis

In abstract you describe 7 covariates but statistical analyses were done only with 6 covariates, missing child’s age.

With 26 prenatal exposed children the study could be underpowered for the ADHD-related symptoms. Did you made a power analysis and could you implement this in the part limitations of the study?

Grammar corrections

i.e. Line 206, “child birth” correct into “child’s birth”

Author Response

Dear Reviewer,

We would like to thank you for taking the time to review our manuscript and for your valuable feedback and suggestions. We appreciate your comments and have carefully considered each of them in revising our manuscript.

We are pleased to inform you that we have made all the changes suggested in your comments. For your convenience, we have attached a file that details all the revisions made to the manuscript.

We hope that you find the revised version of the manuscript to be satisfactory. We are confident that the changes made have strengthened the manuscript and improved its quality.

Thank you once again for your valuable feedback and for your support of our work. We look forward to hearing your thoughts on the revised version of the manuscript.

Best regards,

Karina Jansone

Reviewer 2 Report

The study presents original research on a possible relationship between ADHD and maternal smoking during pregnancy using EEG on school-aged children. It describes useful information and arises a potential new area of future investigation. The article is well-written but requires minor check-ups. 

I recommend taking into consideration the following remarks:

-        I recommend rephrasing the first sentence from the abstract

-        R46: Split this sentence and connect the last part with the following sentence

-        R61: I suggest splitting this sentence to be clearer

-        R65: Rephrase

-        R70: Rephrase

-        R82: I think this should be in another paragraph

-        R86: I recommend describing a little how a normal EEG should look like and which results indicate ADHD before presenting the results from other studies

-        R119: citation template for the 33rd source should be in square brackets

-        R120: missing closing the parenthesis

-        R121: Rephrase to “Between the years 2005 and 2007...”

-     R122: Rephrase to “all of whom had reached a gestational age of at least 30 full weeks”

-    R142: Rephrase “the weekly quantity of cigarettes the father smokes within the household during the pregnancy”

-        R146 - Rephrase to “both ocular and other bodily movements”

-        R152-156 - Rephrase, the filtering part can be made more explicit

-        R157: I think you meant “by”

-        R184 - missing parenthesis

-        R191 - can add a little more details for the 'mgcv' plugin (considering the fact that it bears some statistic importance)

-     R287 – 3.3 subsection – I think all those measurements can be more explicitly presented in a table rather than in a phrase

-        R336: I suggest quoting those “previous studies” you mention

-        R341: I recommend replacing or removing the word “however” from this sentence

-   R347: You mention “some previous studies” yet you quote only one study. I recommend quoting other studies if there are available.

-        R354: Please rephrase or split the sentence

-        R368: Please rephrase

-        R386: I recommend replacing the term “come into play” with something else as you already used this previously too many times. I suggest: involved, implicated.

-        R387: please rephrase

-     Figure 1-the subtitle says:  Association of prenatal tobacco exposure (exposed vs. non-exposed) but only smoking group EEG activity is measured in the forest plot

Author Response

(The authors gave the same response as above.)
